# Single-Cell Transcriptomics Analysis Reveals a Cell Atlas and Cell Communication in Yak Ovary

**DOI:** 10.3390/ijms24031839

**Published:** 2023-01-17

**Authors:** Jie Pei, Lin Xiong, Shaoke Guo, Xingdong Wang, Yongfu La, Min Chu, Chunnian Liang, Ping Yan, Xian Guo

**Affiliations:** 1Key Laboratory of Yak Breeding Engineering of Gansu Province, Lanzhou Institute of Husbandry and Pharmaceutical Sciences, Chinese Academy of Agricultural Sciences, Lanzhou 730050, China; 2Key Laboratory of Animal Genetics and Breeding on Tibetan Plateau, Ministry of Agriculture and Rural Affairs, Lanzhou 730050, China

**Keywords:** yak, ovary, single-cell, cellular atlas, cell communication, theca cell

## Abstract

Yaks (*Bos grunniens*) are the only bovine species that adapt well to the harsh high-altitude environment in the Qinghai-Tibetan plateau. However, the reproductive adaptation to the climate of the high elevation remains to be elucidated. Cell composition and molecular characteristics are the foundation of normal ovary function which determines reproductive performance. So, delineating ovarian characteristics at a cellular molecular level is conducive to elucidating the mechanism underlying the reproductive adaption of yaks. Here, the single-cell RNA-sequencing (scRNA-seq) was employed to depict an atlas containing different cell types with specific molecular signatures in the yak ovary. The cell types were identified on the basis of their specifically expressed genes and biological functions. As a result, a cellular atlas of yak ovary was established successfully containing theca cells, stromal cells, endothelial cells, smooth muscle cells, natural killer cells, macrophages, and proliferating cells. A cell-to-cell communication network between the distinct cell types was constructed. The theca cells were clustered into five subtypes based on their biological functions. Further, CYP11A1 was confirmed as a marker gene for the theca cells by immunofluorescence staining. Our work reveals an ovarian atlas at the cellular molecular level and contributes to providing insights into reproductive adaption in yaks.

## 1. Introduction

More than ninety percent of the global yak (*Bos grunniens*) population lives in the high-altitude regions of the Qinghai-Tibetan Plateau in China [1]. They play a vital role in the production and life of the nomads living in the plateau regions as they provide food, shelter, fuel, and transport [2,3]. Yak is the sole bovid animal that adapts well to the harsh ecological conditions of low oxygen, severe cold, and high ultraviolet radiation on the plateau, which is famous for its high elevations, pristine natural environments, and extreme seasonal variations [4]. Especially, they possess a higher adaptive capacity for reproduction under high-altitude stress compared with other bovid breeds [5]. The mechanism underlying reproductive adaptation remains to be illuminated.

As a crucial reproductive organ, the ovary generates sex hormones to orchestrate female secondary sex characters, harbors developmentally competent and mature oocytes, releases mature oocytes for fertilization, and maintains pregnancy [6,7,8,9]. An ovarian reserve of dormant oocytes within primordial follicles determines female fertility and reproductive lifespan [10,11]. Unlike spermatogenesis starting from sexual maturity, oogenesis begins with fetal life, suspends at the dictyate stage of meiotic prophase I, and restarts in puberty [12,13]. Upon activation, a primordial follicle undergoes several developmental processes to reach maturity, including theca cell generation from stromal cells, granulosa cell proliferation and differentiation, follicle fluid accumulation in the antrum, and oocyte meiosis restart [14,15]. However, the vast majority of ovarian follicles succumb to a degenerative process referred to as atresia which happens at any stage of folliculogenesis [16,17,18]. The primordial follicle number in a pair of yak ovaries ranges from two to three million at birth, about 210,000 of which remain until puberty, while the rest undergo atresia before reaching ovulation. 

Delineating the cellular atlas of the ovary is essential for understanding oogenesis, folliculogenesis, ovulation, and pregnancy maintenance. To date, cellular molecular characteristics within the ovary have been revealed by single-cell RNA-sequencing (scRNA-seq) technology for a few species. Comprehensive cell atlases were constructed for humans [19,20], monkeys [21,22], drosophila [23,24,25], and some fish [26,27,28]. In the reports, ovarian cells of the species were classified into distinct cell types which were characterized based on their specifically expressed genes. However, cell types and their molecular signals that regulate ovary function remain to be uncovered in the yak ovary. Moreover, understanding global cell communication is paramount to elucidate the mechanism of the reproductive adaptation to plateau in yaks. Therefore, a cell atlas and its communicating signals are expected to be delineated for the yak ovary.

In the present study, a comprehensive atlas containing various cell types was successfully constructed through scRNA-seq. The signature genes for each cell type were identified, which may take part in the execution of a cellular function. Cell-to-cell communication between the cell types was constructed based on the highly expressed ligand-receptor pairs. The heterogeneity of the theca cells was analyzed in light of their specifically expressed genes. The study steps first to decipher the molecular characteristics of ovarian cell types and fill the gap between the structure and function of ovarian tissue. The results are very helpful to decipher the mechanism underlying the plateau adaptability of reproduction in yaks.

## 2. Results

### 2.1. Data Quality Control and Principal Component Analysis

After sequencing, scRNA-seq read raw data were converted to a Seurat object through the Cell Ranger analysis pipeline. For quality control of the cellular gene expression, the number of genes (nFeature), the number of unique molecular identifiers (UMIs) (nCount), the expression ratio of hemoglobin genes (percent.HB), the distribution ratio of mitochondrial genes (percent.MT), and the expression ratio of ribosomal genes (percent.Ribosome) were computed and displayed in plots (Appendix A). Cells with unique feature counts of more than 2500 or less than 200 or possessing more than 20% of mitochondrial genes were filtered out. After the quality control, a total of 15,333 single cells expressing 18,402 genes were retained. The scRNA-seq dataset was normalized and centered successfully. The cellular gene expression underwent dimensionality reduction smoothly. The top 18 PCs were considered the optimal PC number to perform downstream clustering.

### 2.2. Cell Clustering and Marker Gene Selecting

A K-nearest neighbor graph was constructed based on Euclidean distance in PCA space. The cell clusters were visualized using uniform manifold approximation and projection (UMAP) in a two-dimensional plot. All the ovarian cells were clustered into 17 cell clusters (Figure 1A) and the top 30 DEGs from each cluster were filtered by their adjusted *p*-values (Wilcoxon rank sum test) (Appendix A). The expression levels and percentages of the representative genes across the different clusters are visualized in the dot matrix (Figure 1B), indicating that each cluster has its own specifically expressed gene except cluster 7. The expression specificity of the top five variable genes of each cluster is shown in a heat map (Figure 1C).

### 2.3. Cell Type Identification

Automatic annotation of the scRNA-seq dataset was obtained by the “SingleR” package (Appendix A). Nevertheless, it seemed that the annotation method was not suitable for the yak ovary because cell types in the annotation could not reflect the truth for some cell clusters. In consideration of this, manual annotation was carried out based on the specifically expressed genes and their biological functions of each cluster (Appendix A). Then, the 17 clusters were annotated successfully into eight cell types including theca cells (one cluster, 4.62%), stromal cells (five clusters, 32.39%), endothelial cells (two clusters, 21.69%), smooth muscle cells (one cluster, 1.69%), natural killer cells (four clusters, 29.97%), macrophages (one cluster, 1.40%), proliferating cells (two clusters, 2.20%), and unknown cells (one cluster, 6.04%) (Figure 2A). Cell numbers in each cell type and each cluster were exhibited in a balloon plot (Figure 2B). The top 30 DEGs from each cell type were filtered by their adjusted *p*-values (Appendix A). Expression scores and percentages of the signature genes across the different cell types are visualized in a dot matrix (Figure 2C), displaying that each cell type has its own specifically expressed gene except the unknown cell. Cell type-specific expression of six representative DEGs per cell type is demonstrated in a heat map (Figure 2D). Most of the DEGs express specifically in their own cell types. The average expression levels of signature genes from each cell type were shown in a heat map (Figure 3A). The expression features of eight cell type-specific marker genes are demonstrated on UMAP plots including CCL5, PECAM1, FBLN1, RPL18A, CYP11A1, RGS5, CENPF, and AIF1 (Figure 3B). As the plots indicated, all the marker genes exhibit expression specificity in the corresponding cell types except RPL18A which expresses highly in all the cell types. In line with the above plots, the violin plots show that the distinct cell types exclusively express their signature genes (Figure 3C).

### 2.4. Gene Ontology and Kyoto Encyclopedia of Genes and Genomes Enrichment for Distinct Cell Types

Gene ontology (GO) and Kyoto encyclopedia of genes and genomes (KEGG) enrichment results were generated based on the DEGs in each cell type (Appendix A). Following the GO enrichment analysis, a bubble plot was produced based on the top five biological processes with the highest enrichment FDRs per cell type, which demonstrates the main biological functions of the eight cell types (Figure 4A). As the plot shows, the DEGs highly expressed in natural killer cells mainly participate in leukocyte cell-cell adhesion, T cell activation, and negative regulation of the immune system, etc. The theca cells are involved in the regulation of wound healing, cholesterol homeostasis, and sterol homeostasis. The proliferating cells tend to be related to nuclear division, organelle fission, and sister chromatid segregation. These cell types play crucial roles in the biological processes coincident with their identities. However, the unknown cells cannot be identified because the pathways in which they participate are primarily enriched in translation and ribosome biogenesis.

### 2.5. Cell Communication between Ovarian Cells

The cell-to-cell interactions between cell types were derived based on the expression of a receptor by a cell type and its corresponding ligand by another cell type (Appendix A). A cell-to-cell interaction network was constructed on the basis of the number of ligands and their receptors (Figure 4B and Appendix A). As the network shows, the theca cell is mainly controlled by proliferating cells, smooth muscle cells, and itself and primarily regulates itself, proliferating cells, and smooth muscle cells. There are cell interactions for theca cells among the cell types, indicating the theca cells play a central regulating role in the cell network in the yak ovary. The top significant ligand-receptor interactions via which the theca cells communicate with the other cell types are demonstrated in a bubble plot (Figure 4C). The bubble plot indicates that there are the most ligand-receptor interactions between theca cells and proliferating cells, suggesting that the proliferating cells are presumably dividing granulosa cells. The theca cells regulate the stromal cells, proliferating cells, and smooth muscle cells mainly by their ligands IGF2, INHA, NECTIN3, and PTN. Oppositely, the theca cells are controlled by the stromal cells, proliferating cells, smooth muscle cells, endothelial cells, and themselves principally through the BMP family.

### 2.6. Potential Functional Heterogeneity of Theca Cells

After cell clustering, the theca cells were clustered into five different subtypes (Figure 5A). The DEGs for each subtype were identified successfully. The expression specificities of five marker genes including SPARCL1, RASSF8, TNN1, ND2, and LAPTM5 are shown in UMAP plots (Figure 5B) and violin plots (Figure 5D). The expression patterns of the top five highly variable DEGs for each subtype are exhibited in a heat map (Figure 5C) and a dot plot (Figure 5E). The average expression levels of the top 10 highly variable DEGs for each subtype are demonstrated in a heat map (Figure 5F). As the diagrams show, subtypes 0, 1, 2, and 4 specifically express SPARCL1, RASSF8, TNN1, and LAPTM5, respectively, while subtype 3 lacks specific DEGs. GO enrichment result for the DEGs of each subtype is obtained (Appendix A) and shown in a bubble plot (Figure 5G). The DEGs in subtype 0 are primarily enriched for the biological processes related to extracellular structure organization and smooth muscle cell proliferation. The enrichment analysis of highly expressed genes in subtype 1 indicates the subtype principally participates in the ribosome metabolic process. The corresponding enrichment for subtype 2 shows the subtype associated with the steroid biosynthetic process and cholesterol metabolic process. The signature genes of subtype 3 are enriched in oxidative phosphorylation, aerobic respiration, and ATP synthesis coupled electron transport. Subtype 4 is mainly involved in T cell activation, leukocyte cell-cell adhesion, and regulation of the immune system.

### 2.7. Pseudotime Expression Patterns for Representative Genes in Theca Cells

The developmental trajectory of the theca cells was constructed to study the transcriptomic pathways that theca cells would take during their differentiation processes. A heat map of the representative genes of the theca cells shows their dynamic expression along the pseudotime, indicating the temporal and progressive dynamics of the representative genes (Figure 6A). Similar trends of the representative genes can be found in the expression patterns of the genes along the pseudotime axis (Figure 6B). As shown in the diagrams, the expression of the marker genes LAPTM5 and ND2 follow a relatively stable pattern. RASSF8 starts with a low level at the initial stage, goes up in the middle stage, and then declines in the late stage. The expression of SPARCL1 begins with a high level, gradually goes down in the middle stage, and then keeps a stable level. TNNI1 seems to be initiated to a transcript at the late stage. The pseudotime of the theca cells is determined by their mapped positions along the principal curves (Figure 6C). We find that this trajectory initiates in subtype 0 of theca cells, further proceeds through subtype 3, 1, and 4 in turn, and ends in subtype 2 with continuous transitions in between.

### 2.8. Immunofluorescence Identification of a Marker Gene of Theca Cells

The immunofluorescence technique was applied to determine whether scRNA-seq accurately localized the specific gene expression to yak theca cells. As the immunofluorescence figure exhibited, CYP11A1 is relatively highly expressed in the theca compared to the granulosa layer and stromal area in an antral follicle (Figure 7A). The magnified immunofluorescence section indicates that the specific expression of CYP11A1 can be successfully detected in the theca cells (Figure 7B). In detail, the expression of the marker gene is mainly located in the cytoplasm of the theca cells.

## 3. Discussion

In the human ovary, various cell types were identified, including oocyte, granulosa cell, theca/stromal cell, endothelial cell, perivascular cell, smooth muscle cell, and immune cell [19,20]. There is a striking similarity of cell types in monkey ovaries to those in human ovaries [21,22]. Furthermore, nearly identical cell types were discovered in mice ovaries, except epithelial cells were identified [29,30]. Our classification of yak ovarian cells is consistent with the ovarian cell types present in the literature, including stromal cells, theca cells, endothelial cells, smooth muscle cells, natural killer cells, and macrophages (Figure 2A). These results indicate that cell types of the ovary are relatively conserved between mammals, implying the consistent biological functions of the mammalian ovaries. In the present study, granulosa cells and oocytes were not identified due to a lack of specific marker genes for the two cell types. Canonical markers from other species were not found in the differential expression in the granulosa cells and oocytes of yaks, reflecting the species specificity of yaks. However, we separated theca cells from the stromal cells through the yak ovary, which contributes to cell type identification within the yak ovary. 

A mammalian ovarian follicle contains an innermost oocyte, surrounding granulosa cells, and thecal cells of outer layers. Theca cells arise during secondary follicle formation and synthesize androgens and progesterone simulated by LH via elevating expression levels of CYP11A1, CYP17A1, and STAR in cattle ovaries [31,32,33]. Although cells with high levels of STAR and CYP17A1 may be considered as the presence of theca cells [34], theca cells were not distinguished from human stromal cells, even in large secondary follicles with a visible theca cell layer [20]. This indicates the close relationship between theca cells and general stromal cells. The molecular markers of theca cells differing from those of stromal cells need to be discovered. In the yak ovary, the specific expression of CYP11A1 was found in theca cells, which can be applied to differentiate theca cells from the stromal cells. Meanwhile, the high intensity of CYP11A1 protein was confirmed by the immunofluorescence validation. The highly expressed CYP11A1 within yak theca cells is coincident with that within cow theca cells, indicating the bovine specificity of theca cells compared with other species.

As the main somatic cell types, theca and granulosa cells are the sites of action and synthesis of a number of sex hormones that are in part responsible for the development of ovarian follicles [35]. Steroidogenesis of granulosa cells in growing follicles is induced by gonadotropin, which plays a crucial role in ovarian function [36]. The steroidogenic genes CYP19A1 and CYP11A1 are remarkedly influenced by FSH in ovarian granulosa cells [37,38]. CYP19A1 as a cytochrome P450 aromatase converts androgens to estrogens in granulosa cells [39,40]. Conversion of cholesterol to pregnenolone is catalyzed by another cytochrome P450 enzyme CYP11A1 located on the matrix side of the inner mitochondrial membrane which is the first and rate-limiting step in the synthesis of the steroid hormones [41]. However, only high expression of CYP11A1 was found in the yak theca cells (Figure 2C,D). The result implies that the conversion of androgens to estrogens is not necessary and the conversion of cholesterol to pregnenolone may be required for the theca cells in the yak ovary during anestrus in yaks. 

Consistent with previous studies [42,43,44] and clinical studies [45] that contradict the existence of oogonial stem cells, we did not find any marker genes for oogonial stem cells, suggesting this was not a cell type in the yak ovary. We cannot exclude the possibility that the oogonial stem cells were so rare that they were not able to be captured by the current technology. Furthermore, oogonial stem cells might have been so sensitive to the tissue treatment or cell dissociation that they were lost during the sample processing. 

Most of the oocytes reside in the ovarian cortex where they are embedded in primordial follicles [46]. Although the ovarian tissue sampled in our study was composed of cortex and medulla, oocytes were not identified among the cell types in the yak ovary. In a canonical process of single-cell sample treatment, the suspended cells usually are subjected to a filtering step with a 40 μm cell strainer to clear larger particles off. The single-cell suspension of the yak ovary underwent the same filtering protocol, the oocytes might have been filtered out due to their relatively huge size. DDX4 resides in the cytoplasm of oocytes and keeps a high expression level during germ-line cell development; it can be considered a marker gene for oogonial stem cells and oocytes [44,47]. The genes GDF9, FIGLA, and ZP3 also demonstrate highly specific expression in the oocytes of humans, monkeys, and mice [19,22,29]. However, the expression specificity of these markers was not found in any cluster of yak ovary. There is a possibility that the yak oocytes possess different feature genes from other species, resulting in no oocytes discovered in the cellular atlas.

A core hub of genes that regulated the cell type-specific markers was revealed by stage-specific regulatory networks at different stages of monkey oocyte development [22]. In humans, four clusters of fetal germ cells of female were identified on the sound basis of exclusively expressed genes, including the mitotic phase, RA signaling-responsive phase, meiotic prophase, and oogenesis phase [48]. Although oocytes within the yak ovary were not identified, we classified the yak theca cells into five subtypes based on the well-characterized marker genes (Figure 5A) and the unique transcriptional landscape of five theca cell subtypes mapped based on their unique scRNA-seq molecular signatures (Figure 5C,F). The five cell subtypes individually participate in extracellular matrix organization, ribosome biogenesis, steroid biosynthesis, oxidative phosphorylation, and immune regulation. These results indicated that theca cells are kept in various states in the ovary during yak anestrus.

Although we revealed the general cellular atlas of the yak ovary, there are still some limitations that should be taken into account. The limited number of yak ovaries selected as the experimental sample might have biased results if the samples do not represent the yak ovary population in anestrus. Studies with larger sample sizes will be better able to confirm our findings in the future. Moreover, the ovary samples collected in April may not be representative of the yak ovary population in anestrus because the status of the yak ovary might have altered subtly during anestrus. Furthermore, the reliability of the signature genes for the cell types identified by our study remains to be confirmed by other experimental methods, particularly of the new-found marker genes. Many in-depth studies are further required to discover the functions of the marker genes for different cell types. In addition, the cell proportions do not necessarily reflect the true cell proportions in yak ovarian tissue, as the cells in our analyses were influenced by sampling sites, tissue handling, and dissociation methods, which could impact different cell types differently.

## 4. Materials and Methods

### 4.1. Ovary Sample Collection

Nonpregnant female yaks, aged from four to five years, kept in anestrus, living in regions with altitudes over 3200 m above sea level in Qinghai Province were recruited as experimental candidates. Ovaries free of anatomical abnormities were removed immediately from yak carcasses after slaughter. The fresh ovaries were rinsed with physiological saline to remove blood. Then, six normal ovaries from different individuals were chosen for the HE staining experiment, three of which were subjected to an immunofluorescence staining test, a further one of which was selected for scRNA-seq analysis. 

### 4.2. Ovary Sample Pretreatment for Single-Cell Dissociation

The ovary was minced into approximately 3 mm cubed pieces with a sterile scalpel blade. Five cubed pieces were obtained and flushed with phosphate-buffered saline (PBS). Each cubed piece was separately transferred into a cryovial containing 1 mL of cell cryopreservation medium (80% DMEM, 10% DMSO and 10% FBS), resuspended adequately, and balanced at room temperature (RT) for 10 min. Then, the cryovials were transferred into a gradient freezing box containing adequate isopropanol. The gradient freezing box was immediately buried in dry ice and underwent gradient freezing for 12 h. Subsequently, the cryovials were transferred into liquid nitrogen for long-term cryopreservation.

### 4.3. Single-Cell Dissociation of Ovary Sample

The ovarian tissue pieces were dissociated for single-cell RNA sequencing. In brief, the tissue pieces frozen in cryopreservation medium were thawed in a water bash at 37 °C, washed twice with cooled RPMI 1640 medium containing 0.04% bovine serum albumin (BSA), and further minced into approximately 0.5 mm^3^ pieces in RPMI 1640 medium containing 0.04% BSA. The pieces were subjected to digestion with enzymes, 1 mg/mL collagenase Type II (Life Technologies, Grand Island, NY, USA) and 0.25% trypsin-EDTA (Life Technologies, USA) on ice overnight. The digestion process was stopped by adding DMEM supplemented with 10% of fetal calf serum (Gibco, Paisley, UK). To collect the dissociated cells, the cell suspension was centrifuged at 160× *g* for 3 min. Next, the dissociated cells were incubated with advanced DMEM/F12 Glutamax (Life Technologies) containing 1% insulin-transferrin-selenium (Life Technologies, USA), 1% penicillin-streptomycin (Life Technologies, USA), and 27 IU/mL RNase-free DNase I (Qiagen, Hilden, Germany) at 37 °C for 1 h. The cells were resuspended in DPBS containing 2% FBS and passed through a 40 μm cell strainer (Corning, NY, USA) to remove remaining cell aggregates.

### 4.4. Single-Cell RNA Library Construction and Sequencing

The dead cells in the cell suspension were removed with a dead cell removal kit (MACS, Milteny Biotec, Bergisch Gladbach, Germany) and single-cell suspensions including high viability cells were obtained. The cell suspension was loaded into chips of a Chromium Next GEM Single Cell 3′ Reagent Kit v3.1 (PN-1000128, 10× Genomics, Pleasanton, CA, USA) and subjected to a Chromium Single Cell Controller (10× Genomics, USA) to generate single-cell GEM. Then, a primer containing Illumina TruSeq Read 1 sequence, a 10× Barcode, a UMI, and a poly-dT sequence was employed to produce barcoded full-length cDNA from poly-adenylated messenger RNA. The GEMs were broken and the pooled fractions were recovered. The barcoded first-strand cDNAs were purified with silane magnetic beads and amplified to generate sufficient mass for library construction. Enzymatic fragmentation and size selection was used to optimize cDNA amplicon size. P5, P7, a sample index, and a TruSeq Read 2 were added to the amplicon via end repair, A-tailing, adaptor ligation, and PCR. Subsequently, the library synthesis and RNA-seq were completed with an Illumina sequencing platform (Novaseq 6000) using a 300 cycles kit (Illumina, San Diego, CA, USA) and paired-end readings of 150 bp were generated. The scRNA-seq data have been deposited in the Gene Expression Omnibus (GEO) with the accession number GSE213989.

### 4.5. Single-Cell Gene Expression Matrix Generation

The raw Chromium scRNA-seq output was processed using the Cell Ranger pipeline provided by 10× Genomics (v2.2.0) and the reads were aligned to yak genome version BosGru3.0 using the STAR aligner [49]. Consequently, a gene expression matrix for the cells with correctly detected cellular barcodes was generated, of which each column represented a valid cell barcode and each row represented a gene.

### 4.6. Quality Control and Normalization for Single-Cell RNA-Sequencing Data

The expression matrix of cells was handled in a standard workflow of R (version 4.1.3, Vienna, Austria) using the R package Seurat version 4.1.0 [50]. After the Cell Ranger filtration based on correctly detected cellular barcodes, cells expressing more than 200 genes and genes expressed in at least three cells were kept for downstream bioinformatics analyses. Percentages of hemoglobin, mitochondrial, and ribosomal features were calculated by the function “PercentageFeatureset”. The nFeature, nCount, percent.HB, percent.MT, and percent.Ribosome were visualized through the function “Vlnplot”. For filtering high-quality cells and excluding cells with extreme values indicating low complexity, duplets, or apoptotic cells, the ovarian cells that expressed gene numbers ranging from 200 to 2500 and possessing a percentage of mitochondrial genes to total genes lower than 20% were retained for further analysis. The remaining data were normalized using the function “NormalizeData” of the R package Seurat. Specifically, the UMI counts of each gene in each cell were divided by the sum of the UMI counts of that gene across all the cells. The obtained values were multiplied by 10,000 and then transformed by a natural logarithm. Subsequently, the top 2000 highly variable genes were identified using the Seurat function “FindVariableFeatures” with the variance-stabilizing transformation method [51]. The obtained variables were centered to a mean of zero and scaled to a standard deviation of one using the function “ScaleData” based on the variable genes.

### 4.7. Dimensionality Reducing, Cell Clustering, and Marker Gene Finding

Dimensionality reduction was carried out on the scaled data via PCA using the function “RunPCA” of Seurat. The number of PCs was determined by the “ScoreJackStraw” and “Elbowplot” function to obtain the optimal condition for clustering. Cell clustering was conducted using the “FindNeighbors” and “FindClusters” functions. The top 18 PCs were chosen as the dimensionality of the dataset. A modularity optimization technique (Louvain algorithm) was employed to cluster the cells, and the most optimal resolution parameter of 0.6 was used after optimizing the parameter ranging from 0.4 to 1.2. The cell clusters were visualized through the nonlinear dimensionality reduction algorithm UMAP using the function “RunUMAP” with a perplexity parameter of 18, giving each datapoint a location on a two-dimensional map. Differentially expressed genes (DEGs) were determined using the “FindAllMarkers” function (Wilcoxon Rank Sum test). In this process, a threshold of base 2 logarithms of fold change was set to 0.25 and minimum percentage of cells expressing specific genes in each cluster was set to 0.25 either. Expression specificity of the DEGs in each cluster was visualized as a heat map and a dot plot generated by the functions “DoHeatmap” and “DotPlot”, respectively.

### 4.8. Gene Ontology and Kyoto Encyclopedia of Genes and Genomes Enrichment and Cell Type Annotation

GO and KEGG enrichment analyses on the top 200 variable DEGs in each cell cluster were implemented by the Bioconductor packages “clusterProfiler” [52] and “org.Bt.eg.db”. The GO enrichment was demonstrated on a bubble plot using the “doplot” function from the R package “ggplot2”. The cell clusters were annotated based on high variable genes in each cluster using “SingleR” package. Subsequently, the clusters were manually annotated into distinct cell types based on the CellMarker dataset, previous relevant reports [53,54], the GO enrichment, and the “SingleR” annotation results. The top 200 variable DEGs in each cell types were also subjected to GO and KEGG enrichment analyses to ensure the reliability of the cell annotation. The annotated cell types were visualized in a two-dimensional UMAP plot by the “DimPlot” function. A dot plot, a heat map, eight feature plots, and eight violin plots were produced by the “DotPlot”, “DoHeatmap”, “FeaturePlot”, and “VlnPlot” functions, respectively, to display the expression specificity of the representative genes. In addition, the average expression level of each signature gene in each cell type was displayed in another heat map. 

### 4.9. Cell Communication Analysis

Cell-cell communication via ligand-receptor interactions was established with the Python package “CellPhoneDB” [55] to explore the potential interaction between different cell types. The cell-cell interplays were inferred based on the expression of known ligand-receptor pairs in the cell types. Only in the situation where a receptor and its ligand were expressed in at least 10% of the corresponding tested cell population was it considered that an interaction existed between two cell types.

### 4.10. Heterogeneity Analysis on Granulosa Cells and Oocytes

The gene expression matrices of granulosa cells and oocytes were extracted from the expression matrix of all cell types. The top 200 highly variable genes were filtered out for each cell type. The obtained variables were centered to a mean of zero and scaled to a standard deviation of one based on the variable genes. Dimensionality reduction was carried out on the scaled data via PCA. The number of PCs was determined to obtain the optimal condition for clustering. Cell clustering was conducted with a resolution of 18 PCs and a parameter set to 0.4. The cell clusters were visualized by UMAP with a perplexity parameter of 18, giving each datapoint a location on a two-dimensional map. DEGs were determined for the two cell types. Expression specificity of the DEGs in each cell type was visualized as a heat map and a dot plot. GO and KEGG enrichment analyses on the top 200 variable DEGs of each cell cluster were implemented. The GO enrichment results were demonstrated on bubble plots. Dot plots, heat maps, feature plots, and violin plots were generated to display the expression specificity of the representative genes. In addition, the average expression level of each signature gene in each cell type was displayed in the other heat maps. The above data analyses were conducted as previously described in the data generation processes of the intact expression matrix. 

### 4.11. Construction of Developmental Trajectory for Granulosa Cells and Oocytes

Pseudotime trajectory analysis was performed to inspect the progression of continuous cell states using the R package “monocle” based on the transcriptional dynamics that occurred in granulosa cells and oocytes [56]. The monocle objects were created from the Seurat objects of granulosa cells and oocytes using the function “newCellDataSet” of the package “monocle” with a lowerDetectionLimit of 0.5. The DDRTree algorithm was applied to visualize the pseudotime trajectory in the reduced dimensional space. Gene expression heat maps over pseudotime were generated with the “plot_pseudotime_heatmap” function with the highly variable genes as input. The pseudotime trajectory plots of granulosa cells and oocytes were built using the function “plot_cell_trajectory” of the package “monocle”. Expression trends of the highly variable genes across pseudotime were visualized using the function “plot_genes_in_pseudotime”.

### 4.12. Immunofluorescence Validation

After being washed with normal saline as previously described, the three yak ovaries were cut into approximately 5 mm cubed pieces. The cubed pieces were immediately rinsed with PBS, fixed overnight in 4% methanol-free paraformaldehyde, transferred to 70% ethanol, and embedded in paraffin using a Shandon Excelsior tissue processor (Thermo Scientific, Altrincham, UK). The paraffin-embedded tissue blocks were sectioned (4 μm thickness) using an RM2065 microtome (Leica Instruments GmbH, Wetzlar, Germany) onto StarFrost slides. For immunostaining, the paraffin sections were deparaffinized in xylene twice, and rehydrated in a series of ethanol baths, ending with distilled water at RT. Antigen retrieval was performed in 0.01 M sodium citrate buffer (pH 6.0) in a steamer for 20 min. After cooling down, the slides were rinsed three times with PBS and blocked with blocking buffer (1% BSA, 0.05% Tween-20 in PBS) at RT for 1 h. The tissue sections were incubated with a primary antibody rabbit anti-TOP2A (1:200, bs-1920R, Bioss, Beijing, China) at 4 °C overnight. On the following day, the sections were incubated with a secondary antibody goat anti-rabbit IgG (1:500, A16118, ThermoFisher, Waltham, MA, USA) at RT for 1 h. The sections were washed with PBS, counterstained with 4′,6-diamidino-2-phenyl-indole (DAPI, Life Technologies, USA), and mounted using ProLong Gold (Life Technologies, USA) in fluorescent mounting media (Dako Agilent, USA). Immunostained slides were scanned with a Pannoramic 250 Flash III digital scanner (3DHISTECH Ltd., Budapest, Hungary) and representative areas were selected for imaging using “Pannoramic Viewer” software (version 2.5.0, 3DHISTECH Ltd.).

## 5. Conclusions

We depicted the first cellular atlas of the yak ovary during anestrus. The specifically expressed genes CCL5, PECAM1, FBLN1, CYP11A1, RGS5, CENPF, and AIF1 can be considered as marker genes for natural killer cells, endothelial cells, stromal cells, theca cells, smooth muscle cells, proliferating cells, and macrophages, respectively, in the yak ovary. A bidirectional network of cell-to-cell communication was constructed on basis of the highly expressed ligand-receptor pairs among the cell types. Especially, we established a network of cell-to-cell communication to demonstrate the interactions between the theca cells and the other cell types. In addition, the cell subtype analysis of the theca cells indicated their heterogeneities in the yak ovary. The results obtained from this study could be beneficial to provide insight into the mechanism underlying the reproductive adaptation to high-altitude environments in yaks. 

## Figures and Tables

**Figure 1 ijms-24-01839-f001:**
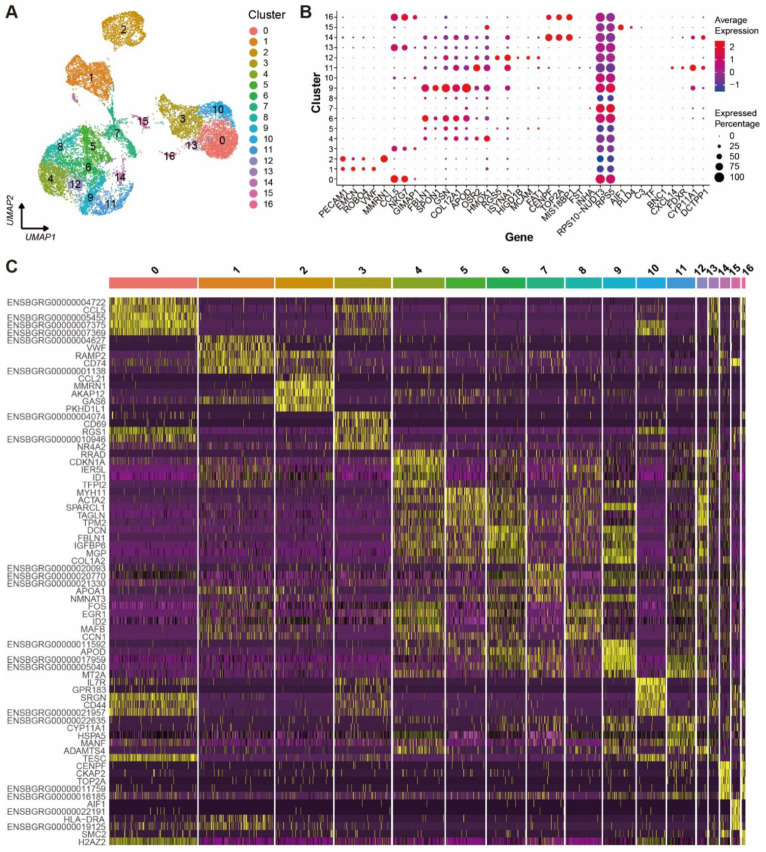
Cell clusters and their signature gene features in yak ovary. (**A**) Uniform manifold approximation and projection (UMAP) scatterplot visualizing various cell clusters. Each point corresponds to a single-cell color-coded according to its cluster membership. (**B**) Dot plot showing expression features of the signature genes selected for each cell cluster among the cell clusters. Gene expression levels from low to high are indicated by a color gradient from blue to red. Percentages of cells expressing specific gene are indicated by size of dot. (**C**) Heat map exhibiting distinct expression patterning of the top five most variable genes for each cluster among the cell clusters. Gene expression levels from low to high are indicated by a color gradient from purple to yellow.

**Figure 2 ijms-24-01839-f002:**
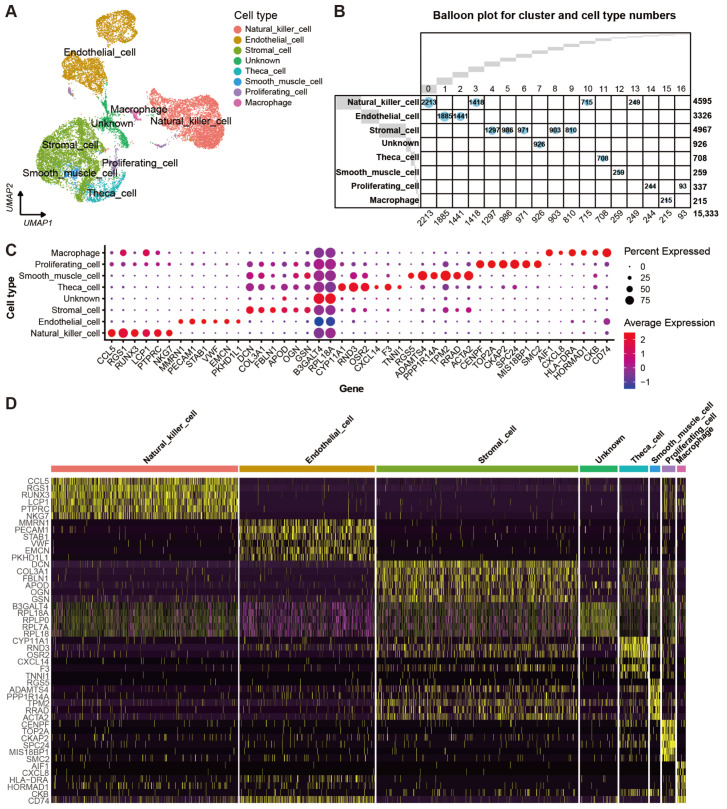
Cell types and their signature gene features in yak ovary. (**A**) Uniform manifold approximation and projection (UMAP) scatterplot visualizing various cell types. Each point corresponds to a single-cell color-coded according to its cell type membership. (**B**) Balloon plot demonstrating amounts of cell clusters and cell types. Each column indicates one cell cluster and corresponding cell number is given at the bottom of the plot. Each row represents one cell type and corresponding cell number is given on the right side of the plot. (**C**) Dot plot showing expression features of the signature genes selected for each cell type among the cell types. Gene expression levels from low to high are indicated by a color gradient from blue to red. Percentages of cells expressing specific genes are indicated by size of dot. (**D**) Heat map exhibiting distinct expression patterning of the top 6 most variable genes for each cell type among the cell types. Gene expression levels from low to high are indicated by a color gradient from purple to yellow.

**Figure 3 ijms-24-01839-f003:**
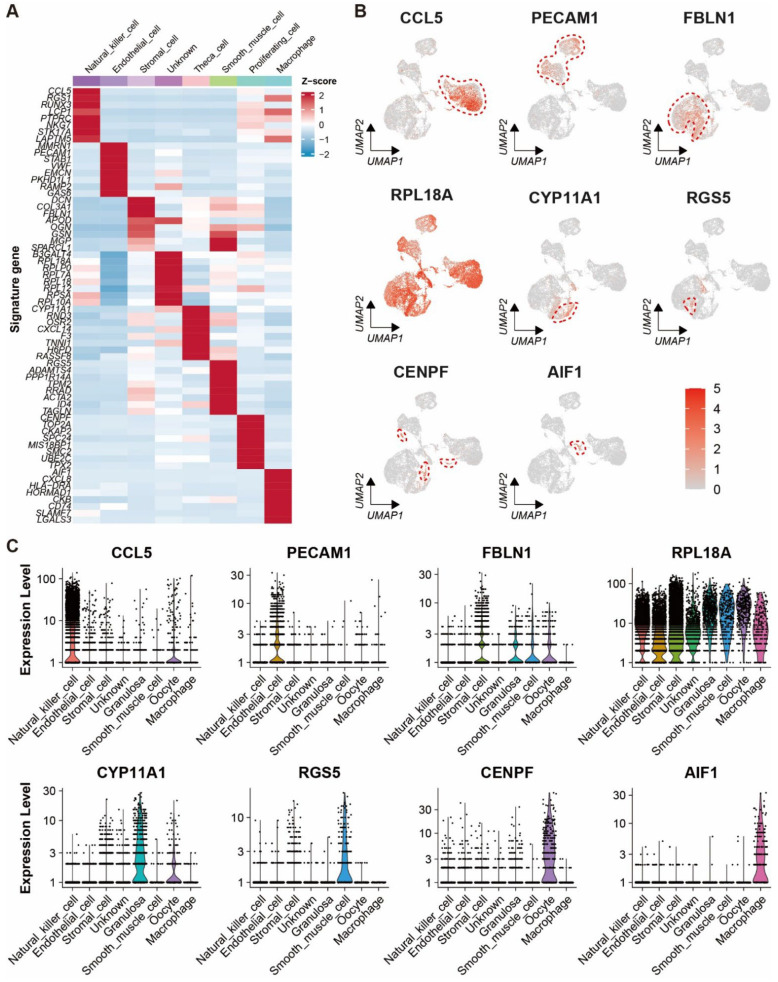
Expression characteristics of marker genes for each cell type in yak ovary. (**A**) Heat map showing an average expression level of each marker gene in each cell type among the cell types. Gene expression levels from low to high are indicated by a color gradient from blue to red. (**B**) Feature plots exhibiting expression specificity of the marker genes across all the ovarian cells. Expression level of each gene from none to high is indicated by a color gradient from light gray to red. Red dashed lines give boundaries of the main cell type of interest. (**C**) Violin plots visualizing expression specificity of the marker genes for each cell type. Expression values of the marker genes were scaled by log-normalization. The vertical coordinate displays expression scores of the marker genes.

**Figure 4 ijms-24-01839-f004:**
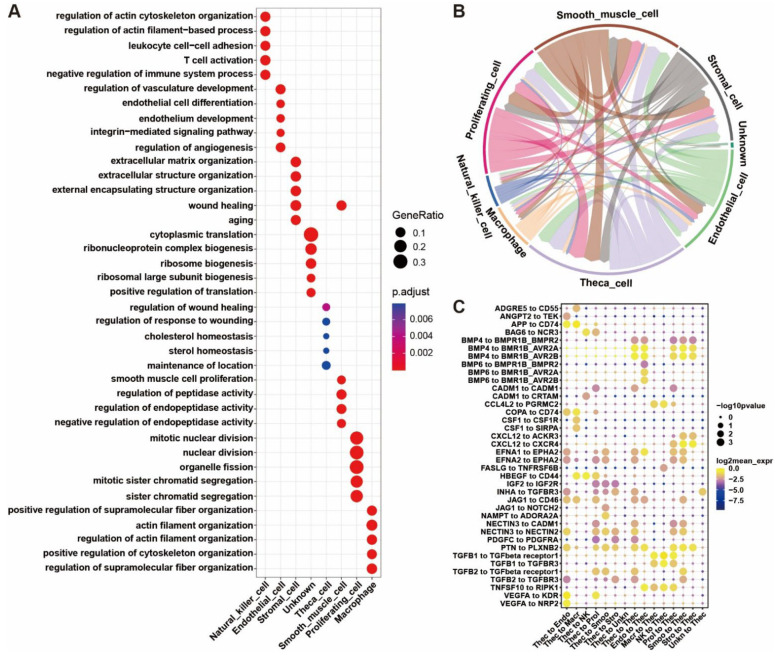
Biological process enrichment for signature genes and cell-to-cell communication between yak ovarian cells. (**A**) Bubble plot showing the top five enriched biological process (BP) terms for each cell type from gene ontology (GO) analysis. Gene ratio in corresponding biological process is indicated by size of bubble. Adjusted *p*-values from low to high are indicated by a color gradient from red to blue. (**B**) Network exhibiting the cell-to-cell communication depending on significant ligand-receptor pairing interactions and the interaction directions. Arrows start from cell types expressing ligands and point to cell types expressing corresponding receptors. (**C**) Bubble plot demonstrating the ligand-receptor interactions via which the theca cells communicate with the other cell types. Endo, endothelial cell; Macr, macrophage; NK, natural killer cell; Prol, proliferating cell; Smoo, smooth muscle cell; Stro, stromal cell; Thec, theca cell; Unkn, unknown cell.

**Figure 5 ijms-24-01839-f005:**
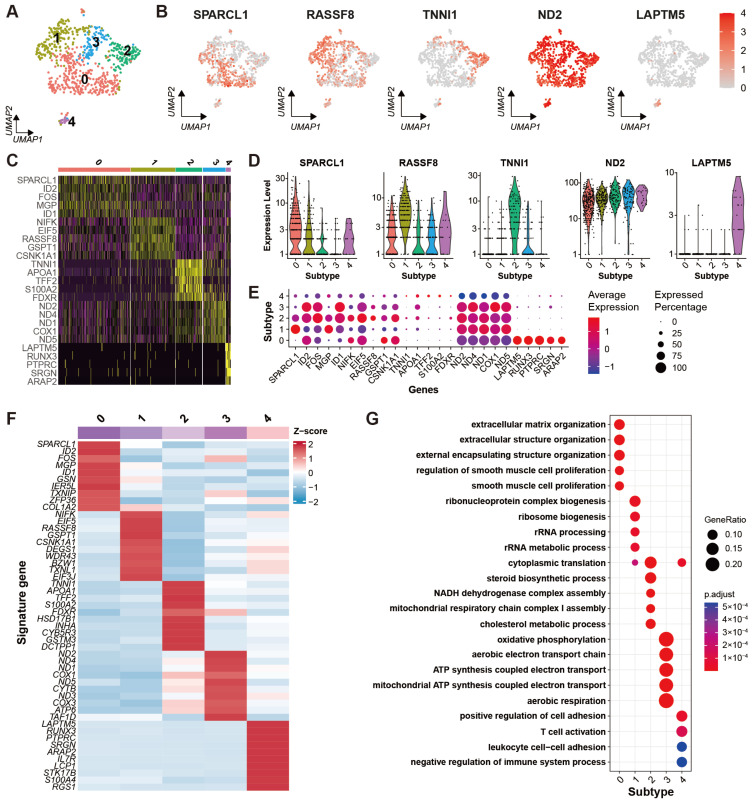
Heterogeneity of theca cells in yak ovary. (**A**) Uniform manifold approximation and projection (UMAP) scatterplot visualizing subtypes of theca cells. Each point corresponds to a single-cell color-coded according to its cell subtype membership. (**B**) Feature plots exhibiting the expression specificity of marker genes across all the theca cells. Expression level of each gene from none to high is indicated by a color gradient from light gray to red. (**C**) Heat map exhibiting distinct expression patterning of the top 5 most variable genes for each theca cell subtype among the cell subtypes. Gene expression levels from low to high are indicated by a color gradient from purple to yellow. (**D**) Violin plots visualizing expression specificity of the marker genes for each theca cell subtype among the cell subtypes. Expression values of the marker genes were scaled by log-normalization. The vertical coordinate displays expression scores of the marker genes. (**E**) Dot plot showing expression features of the signature genes selected for each theca cell subtype among the cell subtypes. Gene expression levels from low to high are indicated by a color gradient from blue to red. Percentages of cells expressing specific genes are indicated by size of dot. (**F**) Heat map showing an average expression level of each marker gene for each theca cell subtype among the cell subtypes. Gene expression levels from low to high are indicated by a color gradient from blue to red. (**G**) Bubble plot showing the top five enriched biological process (BP) terms of each theca cell subtype from gene ontology (GO) analysis. Gene ratio in the corresponding biological process is indicated by size of bubble. Adjusted *p*-values from low to high are indicated by a color gradient from red to blue.

**Figure 6 ijms-24-01839-f006:**
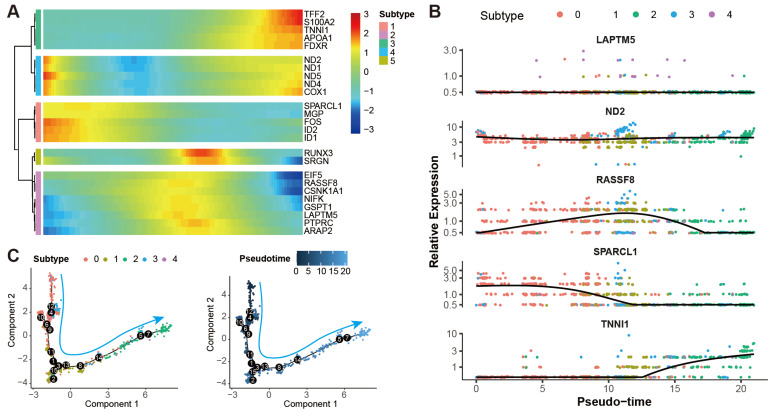
Pseudotime analysis of theca cells in yak ovary. (**A**) Heat map showing gene expression dynamics along pseudotime for theca cells. Expression level of each gene from low to high is indicated by a color gradient from blue to red. (**B**) Scatterplots exhibiting expression tendencies of the signature genes for theca cell subtypes arranged along pseudotime. (**C**) Scatterplots exhibiting the differential trajectories of five theca cell subtypes with pseudotime scale. The numbers in the black bubbles represent cellular states. The cyan arrows stand for the pseudotime directions of the theca cells.

**Figure 7 ijms-24-01839-f007:**
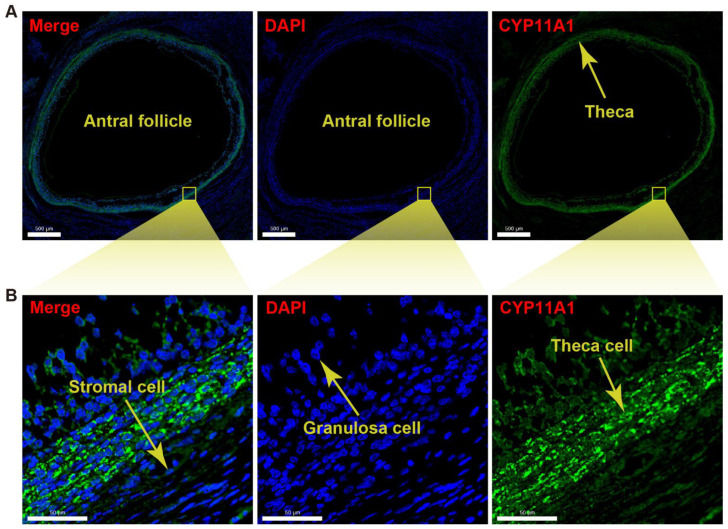
Immunofluorescence detection for CYP11A1 of yak theca cells. (**A**) Immunofluorescence section demonstrating validation of the marker CYP11A1 for a theca layer in an antral follicle. (**B**) Immunofluorescence section exhibiting the magnified theca with high density of CYP11A1. 4′,6-diamidino-2-phenyl-indole (DAPI) was used as nuclear counterstain. The scale bars are 500 μm (**A**) and 50 μm (**B**) in length.

## Data Availability

Data are contained within the article or Appendix A. The scRNA-seq data reported in this publication have been deposited in the Gene Expression Omnibus (GEO) with the accession number GSE213989. https://www.ncbi.nlm.nih.gov/geo/query/acc.cgi?acc=GSE213989 (release date: 1 April 2023).

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
