# Peer review of "Single-Cell Transcriptomics Analysis Reveals a Cell Atlas and Cell Communication in Yak Ovary"

_ijms, 2023, doi:10.3390/ijms24031839_

Round 1
Reviewer 1 Report
In their manuscript, “Single-cell transcriptomics analysis reveals a cell atlas and cell communication in the yak ovary”, Pei et al use single-cell RNA Sequencing and subsequent bioinformatic analysis to define the repertoire of cells present in the yak ovary. An atlas of the yak ovary and analysis of the relationship among various cells is an important goal for understanding how the yak has adapted its reproductive biology to account for the high altitude, harsh environment of their native ecosystem. Based on their analysis, the Authors identified unique heterogeneity within both the granulosa and oocyte compartments, suggesting unique reproductive adaptations present in the yak ovary that have not been observed single-cell analysis of other mammalian ovaries.
Major comments:
Based on the data available to this Reviewer (lacking Supplementary Figures/Tables), it is difficult to assign phenotypes to each Seurat Cluster present, however, it seems very clear from data presented that the Authors have misidentified multiple clusters:
1. Seurat clusters 14 and 16 are unlikely to represent oocytes and the Authors note in line 383 that canonical oocyte marker DDX4 was not expressed in these cells. Many other canonical markers of oocytes exist and none of these are shown either. While the Authors attribute this to the unique biology of the yak ovary, it is more likely that these are subsets of cells from endothelial, stromal and hematopoietic clusters. Previous analyses have shown that the marker TOP2A, along with CENPF (both markers expressed in oocytes in Figure 2D), are enriched in proliferating cells. The mitotically active cells, due to their unique gene expression can often cluster separately from phenotypically similar non-dividing cells. Hence these clusters are likely proliferative subsets of endothelial (upper cluster 14), stromal (lower cluster 14), and hematopoietic (cluster 16) cells. Indeed, in the subanalysis of the putative oocyte group (Figure 6) 5 clusters were identified – cluster 0 expressed PTPRC (CD45) a canonical hematopoietic marker; clusters 1, 2 and 4 (contiguous clusters) expressed OGN and LUM, genes that have been shown to be expressed in stromal cells, and cluster 3 expressed ENG and HES1, endothelial-specific transcripts.
2. Designation of cluster 11 as granulosa cells does not seem well supported by the evidence. Numerous canonical markers of granulosa cells have been identified (e.g. AMH, CYP19A1, CD99, CDH2, FSHR, FOXL2) but expression of these trancripts in the cell population were not shown. Subanalysis of the putative granulosa cells also provides no evidence of expression of these central granulosa-cell specific genes.
Because of the lack of convincing evidence for cluster designation, the analysis and interpretation of the data downstream of this point in the manuscript (Figure 2) is no longer applicable.
The methods used for generating a single cell suspension of yak ovary cells may have contributed to deficiencies in the analysis and cluster designation. According to the Materials/Methods, 3mm cubes were cryopreserved using methods typically applied for single cell suspensions, as opposed to slow-freezing along a controlled temporal gradient, as is the convention for freezing ovarian tissue. Importantly, cryopreserving tissue (using sub-optimal methods) and then thawing and enzymatically dissociating to generate a single-cell suspension, in spite of methods the Authors used to remove dead cells, likely had a negative effect on cell quality. Additionally, without any indication of observed antral follicles within ovarian tissue fragments, the proportion of granulosa/theca cells relative to ovarian stroma is likely to be severely diminished.
Specific comments:
1. The Authors should cite Man et al, Cell Reports 2020 (human) and Morris et al, Elife 2022 (mouse) to fill in their catalogue of ovarian cell atlases.
2. The Authors state in line 103 that the top 5 transcripts for each cluster were selected but clusters 11, 12, 13, 15, and 16 show only 4, 1, 1, 4, and 2 transcripts in Figure 1c. Similarly, in Figure 2d, the top 6 transcripts were selected but only 5 are shown for the Unknown cluster.
3. The language throughout should be edited by a native English speaker to improve clarity.
Reviewer 2 Report
The manuscript by Pei et al. is an interesting application of single cell transcriptomics technology to an important livestock species in an attempt to understand ovarian function. However, there are several issues that must be addressed before this work can be considered for publication.
The Introduction and Discussion are completely lacking any references to existing yak ovary transcriptome and gene expression data. A quick search showed several groups have looked into this already.
On a related note, the scRNA-seq presented was performed on a single ovary. I am quite dismayed that this information was not clearly stated and instead somewhat buried in the methods section. While I understand the expense associated with this technology, it is completely inappropriate to consider this an "atlas" and the title and framing of the paper must be revised accordingly.
At first consideration, the bioinformatics analysis largely appears well done. As there is no mechanism investigated, the results are wholly descriptive, but I appreciate the authors' focus on cell-cell interactions. Subsequent analyses, however, need revision or to be removed entirely. I'm not sure the data presented in Figures 5 and 6 draws appropriate conclusions as there is only one biological sample and few cells sequenced (708 and 337 for GC and oocytes, respectively). Given the sparcity of scRNA-seq data, one cannot infer meaningful measures of diversity from these data and I think this should be removed.
For the trajectory analysis using monocle, it was not clear which starting (and/or ending) state the authors chose. These types of analysis are powerful, but clear experimental questions and appropriate selection of parameters is necessary to get meaningful data. Please explain how this was performed. Additionally, the "subtypes" shown in Figure 7 are not the same as in previous figures (1-5 vs 0-4).
Finally, the two parts of Figure 8 are not related and should probably be separate figures. I would suggest the authors consider removing collagen-integrin interactions from 8A as these tend to overwhelm the other ligand-receptor pairs. For the immunostaining, it is not evident where the oocyte nuclei is located in 8C and that there is a lot of non-specific signal in 8D, suggesting that this gene may not be very oocyte-specific.
Round 2
Reviewer 1 Report
Authors made revisions although the validity and impact of the conclusions remains in question. Significant revision to presentation is also required for the language to be clearly understandable.